# Rapid Detection of Available Nitrogen in Soil by Surface-Enhanced Raman Spectroscopy

**DOI:** 10.3390/ijms231810404

**Published:** 2022-09-08

**Authors:** Ruimiao Qin, Yahui Zhang, Shijie Ren, Pengcheng Nie

**Affiliations:** 1College of Biosystems Engineering and Food Science, Zhejiang University, Hangzhou 310058, China; 2College of Environmental and Resource Sciences, Zhejiang University, Hangzhou 310058, China; 3Key Laboratory of Spectroscopy Sensing, Ministry of Agriculture and Rural Affairs, Hangzhou 310058, China; 4State Key Laboratory of Modern Optical Instrumentation, Zhejiang University, Hangzhou 310027, China

**Keywords:** soil, available nitrogen, ammonium nitrogen, nitrate nitrogen, SERS

## Abstract

Soil-available nitrogen is the main nitrogen source that plants can directly absorb for assimilation. It is of great significance to detect the concentration of soil-available nitrogen in a simple, rapid and reliable method, which is beneficial to guiding agricultural production activities. This study confirmed that Raman spectroscopy is one such approach, especially after surface enhancement; its spectral response is more sensitive. Here, we collected three types of soils (chernozem, loess and laterite) and purchased two kinds of nitrogen fertilizers (ammonium sulfate and sodium nitrate) to determine ammonium nitrogen (NH_4_-N) and nitrate nitrogen (NO_3_-N) in the soil. The spectral data were acquired using a portable Raman spectrometer. Unique Raman characteristic peaks of NH_4_-N and NO_3_-N in different soils were found at 978 cm^−1^ and 1044 cm^−1,^ respectively. Meanwhile, it was found that the enhancement of the Raman spectra by silver nanoparticles (AgNPs) was greater than that of gold nanoparticles (AuNPs). Combined with soil characteristics and nitrogen concentrations, Raman peak data were analyzed by multiple linear regression. The coefficient of determination for the validation ($R_{p}^{2}$) of multiple linear regression prediction models for NH_4_-N and NO_3_-N were 0.976 and 0.937, respectively, which deeply interpreted the quantitative relationship among related physical quantities. Furthermore, all spectral data in the range of 400–2000 cm^−1^ were used to establish the partial least squares (PLS), back-propagation neural network (BPNN) and least squares support vector machine (LSSVM) models for quantification. After cross-validation and comparative analysis, the results showed that LSSVM optimized by particle swarm methodology had the highest accuracy and stability from an overall perspective. For all datasets of particle swarm optimization LSSVM (PSO-LSSVM), the $R_{p}^{2}$ was above 0.99, the root mean square errors of prediction (RMSE_P_) were below 0.15, and the relative prediction deviation (RPD) was above 10. The ultra-portable Raman spectrometer, in combination with scatter-enhanced materials and machine learning algorithms, could be a promising solution for high-efficiency and real-time field detection of soil-available nitrogen.

## 1. Introduction

Soil fertility reflects the ability of soil to coordinate nutrients, water, gas and energy for plant growth and the diverse interactions between biological, chemical and physical properties in the soil [1], which serves as an indispensable indicator of soil quality. Among the various nutrients in the soil, nitrogen (N) is a major essential element of plants [2] and is in great demand. Meanwhile, the native supply of soil N decreases with increasing cropping intensity. By the time visual nutrient-deficiency symptoms are observed, a reduction in yield has occurred. Consequently, N needs to be replenished by fertilization in advance. Virtually all fertilizers are salt. A high electrical conductivity (EC) means a high salt concentration because of more ions in soil conducting more current, and the edge charge associated with soil organic matter (OM) and clays in ion exchange depend on soil solution pH [3].

Soil N exists in various forms, and total soil nitrogen (TN) encompasses them all, signifying the total storage capacity of soil N [4]. TN can be categorized as organic and inorganic nitrogen according to chemical composition. In contrast, soil-available nitrogen is a major nitrogen source for plant uptake and utilization directly, especially inorganic nitrogen [5] in the form of ammonium nitrogen (NH_4_-N) and nitrate nitrogen (NO_3_-N) [2,5,6]. Some organic nitrogen, such as amino acids [7], proteins [8] and amides [9], may be used as a nitrogen source by plants in low N mineralized regions; while cyanide must be converted to NH_4_-N [10] or NO_3_-N [11] before being absorbed. Therefore, we focused on NH_4_-N and NO_3_-N. Their corresponding fertilizers are ammonium nitrogen and nitrate nitrogen, both of which are readily soluble in water. The differences between the two nitrogen fertilizers are that ammonium nitrogen fertilizer may be preferentially used to synthesize amino acids and proteins [12] and it is easily absorbable by soil, whereas, nitrate nitrogen fertilizer is prone to runoff with the movement of water, but it does not reduce calcium (Ca), magnesium (Mg) or potassium (K) uptake like ammonium nitrogen fertilizer [13].

Notably, low nitrogen levels affect plant metabolism and threaten crop yields; excessive nitrogen fertilization lowers ecosystem productivity and causes environmental pollution [14]. Therefore, quantifying the capacity of the soil to supply sufficient nutrients before planting or during the growing season is crucial for optimum plant growth and yield [3]. Analytical chemistry is used to determine the content of available N. Two traditional methods have gained general acceptance: the Kjeldahl [15] and the Dumas [16], which are wet and dry oxidation procedures, respectively. Gas chromatography–mass spectrometry (GC–MS) [17], colorimetric assay [18], electrochemical methods [19], etc. have also been used to quantify nitrogen in the soil. However, the first two methods involve complex pre-treatment steps, which not only consume time and effort but may also increase human error, while the electrochemical method is usually susceptible to temperature and other environmental effects with a shorter service lifespan. Spectroscopy analysis is an increasingly popular technique, due to its simple operation and specific response, including laser-induced breakdown spectroscopy (LIBS) [20], near-infrared spectroscopy (NIRS) [21], hyperspectral imaging (HSI) [22] and the combination of these approaches [23]. Nevertheless, the instruments used in these spectroscopic techniques are relatively expensive, bulky and only available in a laboratory. Moreover, these spectroscopic techniques are mainly based on soil tablets in the solid state, while in this case, the background of the soil sample is complex with many impurities.

Raman scattering is an inelastic scattering process, which is essentially unaffected by water [24]. In the Raman effect, polarized incident light is shifted in frequency by the energy of its characteristic molecular vibrations, leading to a unique spectral output with fingerprint peaks for each molecule [25]. Therefore, Raman spectroscopy is a powerful tool for the characterization of molecules, including nitrogen-containing molecules [26]. Since NH_4_-N and NO_3_-N are easily soluble in water, and plants directly absorb and utilize nitrogen in the soil solution [27], we hypothesized that Raman spectroscopy can generate a resonance response to the chemical bond of available nitrogen in the soil, and filter out impurity information through specific absorption spectra, to realize the qualitative and quantitative detection of soil-available nitrogen. If the hypotheses hold up, we predicted that surface-enhanced Raman spectroscopy (SERS) combined with machine learning algorithms could achieve the detection of soil-available nitrogen with high sensitivity, high precision, good reproducibility and good stability. To verify our hypotheses and predictions, the standard samples were prepared with three soils and two nitrogen fertilizers, and their Raman spectra were recorded by an ultra-portable Raman spectrometer. The specific objectives were to: (1) investigate the ability of SERS to qualitatively identify soil-available nitrogen and the characterization of scattering enhancing nanoparticles; (2) explore the effect of ammonium adsorption on the Raman spectra of soil NH_4_-N; (3) microscopically analyze the principle of specific spectral responses of different soils and nitrogen fertilizers to Raman scattering; (4) establish mathematical regression models at the Raman fingerprint peaks to study the quantitative relationship among the key physical quantities; and (5) verify the feasibility of analyzing full-band SERS spectra by machine learning algorithms.

## 2. Results and Discussion

### 2.1. Physicochemical Properties Analysis of Soil

As shown in Figure 1, soil samples in different latitude and longitude regions cover three typical soils, with Soil_1_, Soil_2_ and Soil_3_ corresponding, respectively, to chernozem, loess and laterite. The intuitive understanding of the inherent characteristics of the three soil samples sets, the T-distributed stochastic neighbor embedding (t−SNE) visualization algorithm was utilized to map the four-dimensional SERS features to two-dimensional space while preserving the internal structure of the original data [28] (Figure 2). It was verified that the differences among the three soil samples were large enough to be representative of universal application.

Regarded as a system of solids, liquids and gases, soil is a complex mixture in the natural environment, providing essential living conditions for plants. Table 1 shows the physicochemical properties of three soil samples, which consisted of two attribute parameters and four main nutrient contents. Soils are all acidic. When ammonium ion (${\mathrm{NH}}_{4}^{+}$) was dissolved in water, two reversible reactions of chemical Equation (1) took place towards the left side to reach the equilibrium state as soon as possible. The wet red litmus papers were placed at the mouth of centrifuge tubes for a few minutes, and the test paper did not obviously turn blue, proving that the loss of ${\mathrm{NH}}_{4}^{+}$ discharged through ammonia gas, was negligible after ammonium sulfate ((NH_4_)_2_SO_4_) was dissolved in DI water. For three soil samples, the pH was in the order of Soil_2_ < Soil_3_ < Soil_1_, and the electrical conductivity was in the order of Soil_3_ < Soil_1_ < Soil_2_. If the minimum value was assigned as one and increased by one in the sequence shown in Table 2, the sum and product of the pH and electrical conductivity conformed to the law of Soil_3_ < Soil_2_ < Soil_1_, which was the same as the order of available nitrogen, potassium and phosphorus concentrations in Table 1. This confirmed that soil pH and electrical conductivity are closely related to the concentration of soil nutrients [3]. The effects of these aspects on soil-available nitrogen have been considered in this research.
(1)$${\mathrm{NH}}_{4}{}^{+}+H_{2}O\overset{}{⇌}{\mathrm{NH}}_{3}\cdot H_{2}O+H^{+}\overset{}{⇌}{\mathrm{NH}}_{3}↑{+H}_{2}O+H^{+}$$

In soil, the nitrogen content is in dynamic equilibrium, as microbial flora not only mobilizes plant nutrients but also induces the nitrogen cycle. Besides soluble nitrogen, insoluble nitrogen contained in the particles of organic matter was converted into plant-available forms through the activities of the bacteria after being mixed into the soil and ingested by earthworms. For instance, nitrite bacteria and nitrate bacteria oxidized nitrogen along the inorganic, organic or combined paths (listed in Figure 3) under aerobic conditions. Under anaerobic conditions, reactive nitrogen (Nr) such as nitrate ion (${\mathrm{NO}}_{3}^{-}$) was converted into its unreactive form by denitrifying bacteria following the pathway below [29]:
(2)$${\mathrm{NO}}_{3}^{-}\to {\mathrm{NO}}_{2}^{-}\to \mathrm{NO}\to N_{2}O\to N_{2}$$

In addition, clays and organic matter have net negative charges [30], causing soil particles to repel negatively charged particles (e.g., ${\mathrm{NO}}_{3}^{-}$); soil attracts positively charged particles (e.g., ${\mathrm{NH}}_{4}^{+}$). The cations bridged organic compounds onto Al-, Fe-, Mn- (oxygen) hydroxides and clay minerals as well [30,31]. This involved a cation-exchange capacity (CEC), which reflected soil fertility and indicated the capacity of soil to retain several nutrients (e.g., ${\mathrm{NH}}_{4}^{+}$) in a plant-available form. The adsorption of ${\mathrm{NH}}_{4}^{+}$ by soil was the result of various factors such as temperature, humidity, pH, soil particle size, ammonium salt concentration [32], etc. According to the results of the ammonium adsorption experiment (Figure 4), Freundlich, Langmuir and Langmuir-Freundlich models [33] were employed to fit isotherm adsorption lines of three soil samples. RSqCOD stands for coefficient of determination, equivalent to squared correlation coefficient (R Squared, R^2^). According to the indicator, Langmuir-Freundlich models had the highest accuracy [34]. Its R^2^ of Soil_1_, Soil_2_ and Soil_3_ was 0.970, 0.994 and 0.986, respectively (three significant digits reserved). Langmuir equations ranked them second, with a gap of less than 0.015 on R^2^. Freundlich models performed the worst because their R^2^ was farthest away from one. The ${\mathrm{NH}}_{4}^{+}$ adsorption isotherms of the three soil samples were different. Accordingly, different soils have different adsorption abilities of ${\mathrm{NH}}_{4}^{+}$ in the same environment because of discrepancy among soil properties owing to diverse composition [35]. As shown in Table 1, that was embodied in the differences in pH and conductivity, as well as organic matter and available potassium, etc. [36] When the same amount of ${\mathrm{NH}}_{4}^{+}$ was adsorbed in the process of maintaining equilibrium, the remaining ${\mathrm{NH}}_{4}^{+}$ concentration in the soil solution was in the order of Soil_1_ < Soil_3_ < Soil_2_.

The soil water containing the nutrients in a dissolved plant-available form was called the soil solution. The plant roots took up nutrients dissolved in the soil solution [27]. Hence, soil nutrients could effectively be plant-available by crops when they are released from the adsorption complex into the soil solution [37]. In the soil, there exists a balance between the nutrients adsorbed on the soil particles and the nutrients released into the soil solution. If this equilibrium is disturbed by nutrient uptake through the plant roots, nutrients are released from the adsorption complex to establish a new equilibrium [27]. As depicted in Figure 5, part of NH_4_-N was attached to the soil, and the other NH_4_-N and NO_3_-N were dissolved into the soil solution. Figure 5 was drawn based on the actual transversal surface of the real objects. In the figure, the blank gap between the soil solution and soil particles represents soil gases. Although ${\mathrm{Ca}}_{2}^{+}$, ${\mathrm{Mg}}_{2}^{+}$ and OH^−^ may participate in the balance adjustment, only ${\mathrm{NH}}_{4}^{+}$ and ${\mathrm{NO}}_{3}^{-}$ were considered here. When plant root hairs touched the soil solution to absorb ${\mathrm{NH}}_{4}^{+}$ and ${\mathrm{NO}}_{3}^{-}$, the concentration of the nitrogen in the soil solution gradually decreased. This meant that the previously mentioned equilibrium was broken, and then transport ① (transport ① in Figure 5) occurred between the soil solution and particles. The nearby soil solution with the original available nitrogen was simultaneously transported to the soil solution with low concentration (transport ② in Figure 5) because of the osmotic effect.

### 2.2. Characterization of Nanosol Substrate

Raman spectra are composed of Raman shifts (expressed in wavenumber units, cm^−1^), and each Raman peak at one Raman shift belongs to a specific chemical bond which means obtained spectra give a vibrational fingerprint of the molecule. Nonetheless, it is very weak in nature because only a very small fraction of the incident photons is inelastically scattered. Thus, the detection of low-abundant molecules in complex media is infeasible [38]. It is necessary to introduce SERS to enhance Raman signals of matters adsorbed on rough noble metal surfaces or nanoparticles by many orders of magnitude [39]. Chemical enhancements (CE) based on the formation of charge-transfer complexes [40] and electromagnetic (EM) field enhancements based on resonance Raman spectroscopy [41] have been used to explain the mechanism of the SERS signal by a factor of 10^3^ and 105–106, respectively. The latter theory is more widely recognized, in which Raman signal enhancement results from surface plasmon polaritons (SPPs).

The Raman enhancements of this study derived from gold nanoparticles (AuNPs) and silver nanoparticles (AgNPs), and they existed in the prepared gold (Figure 6a) and silver sols (Figure 6d), respectively. The nanoparticles exhibited a surface plasmon resonance absorption that is dependent on many factors including the dielectric constants of both the metal and the surface, the interparticle distance, and the shape and size of the particles [42]. The transmission electron microscopy (TEM) images of the nanoparticles were shown in Figure 6b,c and Figure 6e,f, respectively. They were all imaged at 8000 volts, but the images with the 1 μm scale were magnified by a factor of 10,000 for prepared samples, and the images with the 50 nm scale were magnified by a factor of 120,000 times. AuNPs and AgNPs were reasonably dispersed and basically spherical with a diameter of about 50 nm.

The UV-Vis spectra of gold and silver sols are shown in Figure 7. A sharp absorption band was formed at 415 nm (Figure 7, red curve), which was attributed to the surface plasmon response of AgNPs. Likewise, the absorption at 529 nm (Figure 7, blue curve) was attributed to AuNPs.

Scanning electron microscopy (SEM) images of AgNPs are shown in Figure 8a,b to further study the compositional structure of AgNPs. SEM coupled with energy-dispersive spectrometry was adopted for the research. Energy dispersive spectrometry (EDS) is an x-ray characterization technique that records the chosen range to be processed and quantified. In other words, it allows elemental concentrations to be gathered from points, long lines or maps (Figure 8b). AgNPs mainly contain two elements, silver (Ag) and oxygen (O) (Figure 8f). Ag was derived from AgNO_3_ and O came from trisodium citrate dihydrate or DI water. The mass fractions of Ag in the seeds was higher at 28.64 wt% (Table 3). As Figure 8c–e vividly exhibits, when the performance of the two elements is in abundance, Ag aggregates on seeds with a higher density than O, which was also consistent with Figure 8g. Since the absolute scattering intensity contained quantitative information related to the mass and density of AgNPs, the normal absolute intensity increased in proportion to the element.

As shown in Figure 9, the silver nanoparticles enhanced the original Raman spectral intensity by an average of two times, compared to less than 1.3 times for the gold nanoparticles. Without any Raman surface enhanced reagent, the limit of detection (LoD) of NH_4_-N and NO_3_-N were 60 μg/mL and 20 μg/mL, respectively. After adding silver sol, the LoD was reduced to half of the blank enhancement. Though both from citrate reduction, the enhancement of AgNPs was significantly higher than that of AuNPs, probably owing to the stronger electrostatic attraction of AgNPs to ammonium and nitrate molecules. Hence, subsequent data were based on SERS using AgNPs.

### 2.3. Spectral Feature of Soil-Available Nitrogen

In this study, ammonium sulfate ((NH_4_)_2_SO_4_) and sodium nitrate (NaNO_3_) were used to construct a concentration gradient of NH_4_-N and NO_3_-N, respectively. The Raman spectrum peak of soil NH_4_-N took the movement modes of ${\mathrm{NH}}_{4}^{+}$ and ${\mathrm{SO}}_{4}^{2-}$ into account, whereas that of NO_3_-N only considered the movement modes of ${\mathrm{NO}}_{3}^{-}$, because the ionic bond would not be expressed in the Raman spectrum.

${\mathrm{NH}}_{4}^{+}$ is a non-polar ion composed of N-H polar bonds, and its spatial structure is a standard regular tetrahedron (see Figure 10a). The absorption band of NH_4_-N was observed intuitively at 976 cm^−1^, which has been attributed to N-H rocking vibrations [43], as shown in Figure 9a,c,e. Although a minor band can be seen at 451 cm^−1^, resulting from S-O stretching vibration [44], it was not used as a characteristic peak, and its intensity changes with the increase of NH_4_-N concentration was not obvious after all. The van der Waals force was formed between the negatively charged soil colloid and ${\mathrm{NH}}_{4}^{+}$, namely the adsorption of ${\mathrm{NH}}_{4}^{+}$ by the soil colloid. In this process, with heat released, molecular thermal motion was so violent that the hydrogen bonds were easily broken, resulting in the increase in the vibrational frequency of ${\mathrm{NH}}_{4}^{+}$ and the shortening of the wavelength. It was manifested as a shift of 2 cm^−1^ towards the direction of the increasing wavenumber at the Raman peak, which is known as a blue shift. Hence, the actual Raman scatter peak of soil NH_4_-N was 978 cm^−1^. The nitrogen-oxygen bond of ${\mathrm{NO}}_{3}^{-}$ is between single and double bonds, with all four atoms in the same plane (see Figure 10b). The Raman fingerprint peak of soil NO_3_-N was 1044 cm^−1^, which originated from the symmetric stretching vibration of the N-O bond [44,45]. As shown in Figure 11, both NH_4_-N and NO_3_-N in soil had only one vibration mode, so their Raman spectra had unique characteristic peaks. With ten higher concentrations of the three soil samples, the Raman spectral intensities became much stronger, keeping the peak position constant. Accordingly, Raman spectroscopy is a sensitive and selective method.

### 2.4. Model Analysis of Characteristic Peak

With the high signal-to-noise ratio (SNR), the Raman detection device comes with a smoothing filter. Because raw spectra have no apparent sawtooth, the data were directly modelled and analyzed without pre-treatment. The study focused on building regression quantitative models for the datasets corresponding to the two available nitrogen samples in three soil samples, along the route from simple to complex, from univariate to multivariate, and from characteristic peak to full spectrum. The dataset of NH_4_-N and NO_3_-N in Soil_i_ (i = 1,2,3) was nominated as DH_i_ and DO_i_, respectively, and all DH_i_ (i = 1,2,3) was named for DH, all DO_i_ (i = 1,2,3) was similarly named for DO.

The mean Raman peak intensity of the same nitrogen concentration was chosen as the independent variable and the corresponding soil nitrogen concentration as the dependent variable to establish single variable linear regression models. The results of linear fitting are shown in Figure 12 with *y*-axis error bars. The linear regression lines obtained from different soil samples were different. The intercept of the regression line denoted the Raman intensity caused by the adsorbed NH_4_-N or insoluble NO_3_-N in the soil particles when soil solutions had no NH_4_-N or NO_3_-N. For NH_4_-N soil, the intercept of the regression line was in the order of Soil_3_ < Soil_1_ < Soil_2_, consistent with the order of electrical conductivity in three soil samples. For NO_3_-N soil, the intercept of the regression line was in the order of Soil_1_ < Soil_3_ < Soil_2_, opposite to the order of pH in the three soil samples. The slope of the regression line was physically interpreted as the growth rate of Raman intensity with increasing NH_4_-N or NO_3_-N concentrations in the soil solution, equal to the tangent of the angle between the line and the *x*-axis. For NH_4_-N soil, the slope of the regression line was in the order of Soil_1_ < Soil_3_ < Soil_2_, consistent with the order of ${\mathrm{NH}}_{4}^{+}$ concentration remaining in different soil solutions when the same amount of ${\mathrm{NH}}_{4}^{+}$ was adsorbed. The processed results of DH_i_ (i = 1,2,3) were in agreement with that of ${\mathrm{NH}}_{4}^{+}$ adsorption experiments. For soil NO_3_-N, the slope of the regression line was in the order of Soil_3_ < Soil_2_ < Soil_1_, the same as the order of the sum or product of pH and electrical conductivity in the three soil samples. Overall, both pH and electrical conductivity may affect soil nutrient content. Additionally, ${\mathrm{NH}}_{4}^{+}$ adsorption by soil particles must be considered for NH_4_-N. The performance of the single variable linear regression model is shown in Table 4. In general, the coefficient of determination for validation ($R_{p}^{2}$) of the six datasets were all more than 0.96, and the regularized root mean square error (RMSE_r_)was still within the acceptable range, indicating that the models had superior predictive ability. For the three soil samples, the regularized relative prediction deviation (RPD_r_) of NH_4_-N was above 3, while the RPD_r_ of NO_3_-N was below 3, denoting that the single variable linear models of NH_4_-N were more stable and reliable.

Furthermore, multiple linear regression (MLR) models were developed based on the previous inference. The difference from single variable linear regression models was the introduction of soil parameter. The soil factor of NH_4_-N was the product of *q_m_*, *k_L_*, pH and electrical conductivity, while that of NO_3_-N corresponded to the product of the latter two parameter values. Here, the intensities of all Raman characteristic peaks at the same concentration were used instead of their average values. The linear regression results were shown in Figure 13, where three-dimensional (3D) images were obtained. Each variable represents a dimension. To facilitate the observation and comparison, colorful 3D surface grids were also plotted. As shown in Table 5, the model of NH_4_-N was still the better performing model with its $R_{p}^{2}$ of 0.976 and RPD_r_ of 6.43. Moreover, the overall performance of the multiple linear regression of NO_3_-N improved over the univariate linear regression model, because the RPD_r_ of MLR grew to above three while keeping $R_{p}^{2}$ and RMSE_r_ from going in a bad direction as much as possible. The validity of the models was demonstrated, and the solution was provided for a deeper interpretation of the intrinsic connection between Raman spectroscopy and soil-available nitrogen.

### 2.5. Model Analysis of Full Band

The target band of the study was limited to 400–2000 cm^−1^. Partial least squares (PLS), back-propagation neural network (BPNN) and least squares support vector machine (LSSVM) were selected for quantitative analysis in this whole band range, in which intensity at different Raman shifts was taken as the independent variable and the nitrogen concentration was the dependent variable. For cross validation of the three models, the calibration set and the validation set were divided according to the ratio of 4:1 using the SPXY (sample set partitioning based on joint x-y distances) method [46], signifying that 80 out of every 100 samples were used for training the model with the remaining 20 samples used for testing. We constructed the BPNN with a hidden layer of 10 neural nodes and the LSSVM was optimized by particle swarm optimization (PSO) which determined the γ (gam) of 300 for each dataset and σ^2^ (sig2) varied considerably from 12.8 to 200.

The performance of all data points in the NH_4_-N and NO_3_-N datasets on calibration and validation are shown in Figure 14 and Figure 15, and the comparison of their actual and predicted values are shown in Figure 16 and Figure 17. As can be seen from Table 6, the $R_{p}^{2}$ is lower than the coefficient of determination for calibration ($R_{c}^{2}$) in any case. For three models, their $R_{c}^{2}$ and $R_{p}^{2}$ were no less than 0.989 and 0.935, respectively, the root mean square error of calibration (RMSE_c_) and the root mean square error of prediction (RMSE_P_) were no more than 0.321, and the relative prediction deviation (RPD) was at least 3.783. This suggested that PLS, BPNN and LSSVM could be used for the quantification of different soil-available nitrogen, because of sufficiently high accuracy and stability of the predicted results. If improvement is required, combining visual graphs, it is not difficult to find that LSSVM performs the best among all models, especially for the DO_2_ dataset where the $R_{p}^{2}$ is almost one ($R_{p}^{2}$ = 0.9997), and the RPDs were always the largest except for the DH_2_ dataset where the RPDs were slightly smaller than the PLSes. PLS performed second, which was inferior to BPNN only on DH_3_ and DO_2_ datasets. The deviations of the predicted values of BPNN from the actual values were relatively large, as shown in Figure 17b,h.

### 2.6. Prospects for Application and Implementation

At present, the detection of soil is likely to be carried out under laboratory conditions. After samples are transferred from the field to the laboratory, losses caused by environmental changes and transport leakage is usually ignored, which ought not to happen in real agricultural work. As a result, a convenient and efficient method is urgently need to support the in situ detection of soil nutrients, as well as a low-cost and portable device. For this purpose, this study found the feasibility of the ultra-portable Raman spectrometer for quantitative determination of soil-available nitrogen, which included: (1) sensitive response to the target object and high accuracy of detection results; (2) inexpensive acquisition and maintenance costs unlike LIBS, NIR HIS, etc.; (3) suitcase packaging design for easy portability; and (4) uncomplicated sample preparation process, avoiding solid-state tableting through liquid phase detection to save time and effort. In fact, it was possible to develop an automatic sample preparation module for the Raman spectrometer, and then perform secondary integration with the packaged computational model. After one-key startup, the reproducible results could be quickly obtained through mechanical self-processing and computerized self-operation to realize real-time monitoring. The integration of a global position system (GPS) and navigation technology made it possible to know the soil fertility under each plant, so as to replenish the appropriate amount of fertilizer in time, protect environmental resources, optimize return on investment and ultimately achieve precision agriculture.

## 3. Materials and Methods

### 3.1. Materials and Apparatus

In this study, the topsoil (0–20 cm) in three areas of China were designated as Soil_1_, Soil_2_ and Soil_3_, respectively. As shown in Figure 1, Soil_1_ was collected from a Jiamusi City suburb, Heilongjiang Province, China (130°16′12″ E, 46°42′47″ N); Soil_2_ was sampled from Jiujiang City, Jiangxi Province, China (115°56′3″ E, 29°35′23″ N); and Soil_3_ was from Xundian County, Yunnan Province, China (103°07′29″ E, 25°26′22″ N). The four components of the three soil samples were detected: organic matter determined by potassium dichromate oxidation-spectrophotometry, available nitrogen determined by alkaline hydrolysis diffusion method, available phosphorus determined by sodium bicarbonate leaching-molybdenum antimony anti-spectrophotometry, and available potassium determined by flame atomic absorption spectrometry.

The reagents included: ammonium sulfate (AR, ≥99.0%), sodium nitrate (AR, ≥99.0%), silver nitrate (AR, ≥99.8%), phenolphthalein (Ind) and ethanol (AR, ≥99.7%) purchased from Sinopharm Group Chemical Reagent Co., Ltd. (Shanghai, China); chloroauric acid hydrate (AR, 99%, Au:50%, Shanghai Titan Scientific Co., Ltd., Shanghai, China); trisodium citrate dihydrate (AR, 99%, Shandong Xiya Chemical Technology Co., Ltd., Linyi, China); sodium hydroxide (AR, 96.0%, Chengdu Aikeda Chemical Reagent Co., Ltd., Chengdu, China); formaldehyde solution (37–40 wt% Tianjin Bodi Chemical Co., Ltd., Tianjin, China).

In addition to the above materials, the apparatus required is as follows: RmTracer-200-HS portable Raman spectrometer combined with a 785 nm excitation wavelength diode-stabilized stimulator (OptoTrace Technologies, Inc., Silicon Valley, CA, USA) for spectral data acquisition; TU-19 ultraviolet-visible spectrometer (UVS, Beijing Purkinje General Instrument Co., Ltd., Beijing, China) for nanoparticles characterization; JEM-1200EX transmission electron microscope (TEM, JEOL Ltd., Tokyo, Japan) and GEMINI 300 field emission scanning electron microscope (FESEM, ZEISS Ltd., Jena, Germany) for observing the morphology of the nanosol, and the latter equipped with ESPRIT 2.0 (Bruker Ltd., Berlin, Germany) for qualitative analysis; SZCL magnetic stirrer with heating mantle (Zhengzhou Biaohe Instrument Co., Ltd., Zhengzhou, China) for controlling nanosol synthesis; pH meter (LEICI PHSJ-4ApH, Shanghai Kuosi Electronics Co., Ltd., Shanghai, China) for measuring soil pH; conductivity meter (LEICI DDSJ-318, Shanghai Kuosi Electronics Co., Ltd., Shanghai, China) for measuring soil electrical conductivity; laboratory oven (DHG-9030A, Shanghai Yiheng Scientific Instruments Co., Ltd., Shanghai, China); 100-mesh sieve (Zhejiang Shangyu Yarn Screen Factory, Shaoxing, China) with 150 μm aperture size; FA1004B electronic balance (Shanghai Shangping Instrument Co., Ltd., Shanghai, China) with the precision of 0.1 mg; ultrapure water system (Nanjing Yipu Yida Technology Development Co., Ltd., Nanjing, China) for deionized water (DI water); Vortex-Genie 2/2T vortex mixer (Shanghai Ling Early Environmental Protection Instrument Co., Ltd., Shanghai, China); litmus paper (Sinopharm Group Chemical Reagent Co., Ltd., Shanghai, China) for testing alkaline substances; laboratory consumables such as 0.22 μm disposable syringe filters, 9 cm diameter filter papers and 2 mL quartz bottles.

### 3.2. Ammonium Adsorption Experiment

The purpose of this experiment was to investigate the adsorption of ${\mathrm{NH}}_{4}^{+}$ in soil solutions and to determine the effect of different concentrations of ammonium salts on the adsorption balance. First, we used 40 mL DI water to dissolve different masses of (NH_4_)_2_SO_4_ for preparing six ammonium sulfate solutions with concentration gradients of 5, 20, 37.5, 75, 125, and 200 g/L, respectively. Then an 8 g soil sample was mixed with each (NH_4_)_2_SO_4_ solution and left to stand until the stratification of soil liquid was stable. Next, all the liquid was filtered with filter paper. Six gradients of three soil samples were performed as above to obtain 18 filtrates. It was necessary to measure the ${\mathrm{NH}}_{4}^{+}$ content remaining in the filtrate to confirm the concentration of adsorbed ${\mathrm{NH}}_{4}^{+}$ by the soil sample.

Since ammonium salt reacts with formaldehyde (HCHO) to generate hexamethylenetetramine salt and acid [47]:(3)$$4{\mathrm{NH}}_{4}^{+}+6\mathrm{HCHO}\to {\left({\mathrm{CH}}_{2}\right)}_{6}N_{4}H^{+}+6H_{2}O+3H^{+}$$

Phenolphthalein solution is colorless when mixed with acid, and turns red when exposed to alkali. In this experiment, base titration was used. The following chemical reaction occurs during the titration:(CH_2_)_6_N_4_H^+^ + 3H^+^ + 4OH^−^→H_2_O + (CH_2_)_6_N_4_(4)

Since phenolphthalein is insoluble in water, the phenolphthalein solution used ethanol as the solvent. In addition, since formaldehyde solution may contain formic acid, which must be neutralized in advance. A 3 mL phenolphthalein solution was added dropwise to 100 mL formaldehyde solution, and 0.1 mol/L standard NaOH solution was slowly added until its color became faint red, then we diluted the solution with DI water to obtain 20 wt% formaldehyde solution. After preparing these materials, we added neutralized formaldehyde solution to each filtrate, stirred and let stand for a Reaction (3). Next, we added 0.2 wt% phenolphthalein solution, and then the standard NaOH solution was added until the solution appeared pink and did not fade for at least 30 s. According to chemical Equation (4), the concentration of ${\mathrm{NH}}_{4}^{+}$ remaining in the filtrate after adsorption equilibrium was determined by the volume of standard NaOH solution used. All the above operating procedures were conducted under a constant humidity and temperature (25 °C).

Freundlich and Langmuir models were extensively used for describing solute sorption onto the surface of a solid. Herein, the Freundlich model assumed that the adsorption sites of ${\mathrm{NH}}_{4}^{+}$ (adsorbent) were uniformly distributed on the surface of the soil (adsorbent), whereas Langmuir was based on a continuous monolayer of ${\mathrm{NH}}_{4}^{+}$ covering a homogeneous flat soil surface. The Langmuir-Freundlich model, combining the two models, was able to simulate both homogeneous and heterogeneous distributions of adsorbents at both high and low concentrations [48], with higher accuracy and wider applicability generally. Three adsorption isotherm models were used to fit and analyze the experimental data.

Freundlich equation:(5)$$q_{e}=k_{F}c_{e}{}^{n}$$

Langmuir equation:(6)$$q_{e}=\frac{q_{m}k_{L}c_{e}}{1+k_{L}c_{e}}$$

Langmuir-Freundlich equation was a power function based on the assumption of continuously distributed affinity coefficients [33]:(7)$$q_{e}=\frac{q_{m}k_{L}c_{e}{}^{n}}{1+k_{L}c_{e}{}^{n}}$$
where *q**_e_* was the mass ratio of adsorbed ${\mathrm{NH}}_{4}^{+}$ and soil sample at the adsorption equilibrium, *c_e_* was the concentration of the remaining ${\mathrm{NH}}_{4}^{+}$ in solution at the adsorption equilibrium, *q_m_* was the maximum adsorption capacity, *n* was the heterogeneous factor, *k_F_* was the Freundlich constant representing the adsorption capacity of the soil sample, and *k_L_* was the Langmuir equilibrium constant representing the ratio of adsorption to desorption rate.

### 3.3. Nanosol Substrate Synthesis

The citrate thermal reduction method was widely used for colloids [49,50]. In the experiment, silver sol and gold sol were produced by using 1% *w*/*v* trisodium citrate dihydrate (C_6_H_5_Na_3_O_7_·2H_2_O) to reduce 0.01% *w*/*v* chloroauric acid hydrate (HAuCl_4_·xH_2_O) and 0.01% *w*/*v* silver nitrate (AgNO_3_), respectively, with DI water as solvent. During the reactions, it was heated to 360 °C by a magnetic stirrer with a heating mantle, temperature was maintained and the mixture stirred until the silver sol turned into a sufficiently stable green-yellow (Figure 6a) or the gold sol turned into a sufficiently stable dark red (Figure 6d). These color changes resulted from the decrease in metallic particle size over the course of reactions, especially the colloidal gold solution from black to purple to blue to dark red [49]. Finally, we stored two nanosols in the dark at 4 °C, after cooling to room temperature.

### 3.4. Assay Sample Preparation

The fresh soil was first dried in the laboratory oven at 40 °C, then ground and sieved through 100-mesh sifter to gain soil samples. Representative nitrogen fertilizer standards, namely ammonium sulfate ((NH_4_)_2_SO_4_) and sodium nitrate (NaNO_3_), were selected as surrogate available nitrogen and fully dissolved in DI water. Subsequently, the solutions were blended with 10 g soil samples in a mass ratio of 5:1. After adequately vortexing, we let stand for a few minutes (Figure 18). Finally, the supernatant was filtered by the matching syringes and filters to obtain the liquid sample.

Six sample sets came from the mixed solution of the three soil samples and two nitrogen fertilizers, as shown in Table 7. Each sample solution contained ten concentration gradients, and each concentration gradient set up 10 parallel samples. Therefore, a total of 100 (10 × 10 = 100) samples were set for each sample set.

### 3.5. Raman Spectra Acquisition

After x-axis calibration, a portable Raman spectrometer was used for spectral data of the liquid samples. The device was configured with some parameters, such as 200 mW power, range 100–3300 cm^−1^, 2 cm^−1^ optical resolution, 10 s integration, 1 filter smoothing parameter and 2 times scan to find the average. If 1 mL liquid sample in the quartz bottle was directly put into the liquid pool and scanned by Raman spectrometer with 785 nm excitation (Figure 19), the pure Raman spectral data was obtained; but if the silver sol or gold sol prepared was added to the liquid sample before detection, the Raman spectra would be enhanced. The volume ratio of additional silver sol to a liquid sample was 1:200, and it was 1:100 for the gold sol. The target Raman shift was from the range of 400–2000 cm^−1^, so the data of one sample actually consisted of 801 consecutive discrete point values.

### 3.6. Spectral Data Modeling

Chemometry was used throughout the entire study. This technology has been widely applied to solve various qualitative and quantitative problems [51], not confined to the field of chemistry and requires interdisciplinary knowledge [52], such as a professional background to dissect mechanisms, statistical analysis to predict trends, computer programming to process efficiently and mathematical modeling to standardize problem [53]. In this study, the following methods or models were used for modeling and visualization.

Multiple linear regression (MLR) uses multiple independent variables given to analyze the relationship between them and a single dependent variable, which assigns actual explanatory significance to each variable and predicts the outcome of the response variable [54]. The MLR equation:(8)$$y=a_{0}+a_{1}x_{1}+a_{2}x_{2}+\cdot \cdot \cdot +a_{n}x_{n}(n\in Z^{+})$$
where *a* was the regression coefficient, *x* was the independent variable, *y* was the dependent variable, and *n* was the number of items, particularly it was the unary linear regression when *n* = 1. They were used in the data analysis of Raman peaks.

Partial least squares (PLS) regression extracts principal components of the independent variable and the dependent variable, projects them into a new space through the conjugate gradient method, and establishes a linear regression model, avoiding the collinearity problem [55].

Back-propagation neural network (BPNN) is a self-organized machine learning algorithm of the neural network that feeds back the error and outputs forward propagation. In supervised training, the weights of the connections in the network are repeatedly adjusted to minimize the measurement of the difference between the actual output vector and the desired output vector [56].

Least squares support vector machine (LSSVM) is the least squares optimized version of support-vector machines (SVM); it uses equality constraints by solving linear equations instead of quadratic programming, and maps the data into a higher dimensional input space to construct an optimal separating hyperplane for classification and regression [57]. Training and simulation of each output by separately passing Gaussian radial basis function (RBF) kernel as a column vector. LSSVM needs two tuning parameters, *γ* (gam) and *σ^2^* (sig2). gam is the regularization parameter, determining the trade-off between the training error minimization and smoothness, and sig2 is the squared bandwidth in the RBF function [58]. The selection of these parameters plays an important role in the performance of LSSVM. In this study, a particle swarm methodology was employed to optimize LSSVM, called the particle swarm optimization-LSSVM (PSO-LSSVM). It was initialized as a group of random particles, and then found the optimal solution through iteration. In each iteration, the particle updated itself by tracking *pbest_i_* and *gbest_i_* (*i* = 1, 2, …, N, and N was the swarm size). *pbest_i_* denoted the best position for the current particle, and *gbest_i_* denoted the global best position. After finding these two optimal values, the particle updated its velocity *v_i_* and current position *presentx_i_* by the following formula [59]:(9)$$presentx_{i}=presentx_{i}+v_{i}$$
(10)$$v_{i}=\omega \times v_{i}+c_{1}\times rand\left(\;\right)\times \left(pbest_{i}-presentx_{i}\right)+c_{2}\times rand\left(\;\right)\times \left(gbest_{i}-presentx_{i}\right)$$
where inertia weight ω was used to balance the capabilities of global exploration and local exploration; *c_1_* and *c_2_* were two acceleration constants regulating the relative velocities concerning the best local and global positions (*c*_1_ = *c*_2_ = 2, generally); *rand* ( ) was a random variable in the open interval (0, 1)*; γ* and *σ^2^* corresponded to the column vectors of *presentx_i_*.

T-distributed stochastic neighbor embedding (t−SNE) visualizes high-dimensional data into a two- or three-dimensional map, with the capability of retaining the local structure of the data and revealing some important global structures [28]. It constructs a probability distribution for high-dimensional objects in such a way that the more similar the data points, the higher the probability of the assignment, then converts the calculated similarity distances to joint probabilities according to the t-distribution. In the low-dimensional map, t−SNE defines a similar probability distribution and minimizes the Kullback–Leibler divergence between two distributions.

Almost all data processing and analysis were based on MATLAB R2021a (Mathworks Inc., Natick, MA, USA) and Origin 2021 (OriginLab Ltd., Northampton, MA, USA).

### 3.7. Model Performance Evaluation

The evaluation indexes of the regression model adopted in this study were the coefficient of determination (R^2^), the root mean square error (RMSE) and the relative prediction deviation (RPD). R^2^ reflected the correlation between the actual value and the predicted value, RMSE referred to the errors of both and RPD evaluated the stability performance of the model. More specifically, the subscript c and p indicated that the data belonged to the calibration and validation sets, respectively. For example, $R_{c}^{2}$ and $R_{p}^{2}$ represent the coefficient of the determination of the calibration set and the validation set, respectively, in the same way as RMSE_c_ and RMSE_p_. Since Raman spectral intensities were in the thousands, RMSE and RPD were regularized, which could be identified as RMSE_r_ and RPD_r_, respectively. In general, the closer the R^2^ was to 1, the closer the RMSE was to 0 and the larger RPD, the better the performance of the model. Furthermore, the model had the value of practical application when its R^2^ was greater than 0.9 and RPD was greater than 3. The different usages of the dataset do not limit these judgements. For example, $R_{c}^{2}$ and $R_{p}^{2}$ were consistent with the requirements of R^2^, and other indexes are similar.

## 4. Conclusions

This study was the first to use SERS to detect available nitrogen in different soils. We found that the NH_4_-N and NO_3_-N of the three soil samples had unique Raman characteristic peaks at 978 cm^−1^ and 1044 cm^−1^, respectively, and the enhancement of AgNPs in sol was higher than that of AuNPs. When using silver sol to enhance Raman scattering, the LoD of NH_4_-N and NO_3_-N soils were 30 μg/mL and 10 μg/mL, respectively. Although it has already reached the trace level, the magnitude of enhancement was far less than thousands, otherwise the LoD could be down to the ppb (even ppt) level, which will be one of the directions for ongoing research. In addition, it was proved by experiments that ${\mathrm{NH}}_{4}^{+}$ was indeed adsorbed by soil, and ${\mathrm{NH}}_{4}^{+}$ adsorption isotherms of three soils were determined by using Freundlich, Langmuir and combined models, indicating that the Langmuir-Freundlich model provided the best fit, up to 0.994. The less ${\mathrm{NH}}_{4}^{+}$ adsorbed to the soil, the more NH4-N remaining in the solution and the higher the Raman intensity.

The regression analysis of the peaks demonstrated a good linear response between nitrogen concentration and the Raman intensity, with a minimum $R_{p}^{2}$ of 0.963 for six datasets. Based on the adsorption equation combined with the pH and electrical conductivity of the soils to set up the soil factor, 3D linear regression models were established to interpret the internal relationship between Raman spectroscopy and available nitrogen in different soils. Moreover, machine learning algorithms were employed to develop regression models between the soil-available nitrogen concentration and Raman spectral intensity in the range of 400–2000 cm^−1^. General comparative analysis revealed that PSO-LSSVM outperformed PLS, while PLS outperformed BPNN. The worst performance of PSO-LSSVM on soil NH_4_-N was that the $R_{p}^{2}$ was 0.993, the RMSE_P_ was 0.102 and the RPD was 10.42; for NO_3_-N, the $R_{p}^{2}$ was 0.998, the RMSE_P_ was 0.046 and the RPD was 21.66. The prediction effect of the full band analysis based on machine learning was better than that of fingerprint peak analysis based on linear regression.

Overall, SERS combined with machine learning algorithms enabled rapid and accurate assessments of soil nitrogen availability, as an alternative to other spectroscopy techniques. The secondary development of the portable Raman spectrometer has the potential to achieve efficient real-time, in situ monitoring of soil nutrients, thus taking one step closer to scientific fertilization in precision agriculture.

## Figures and Tables

**Figure 1 ijms-23-10404-f001:**
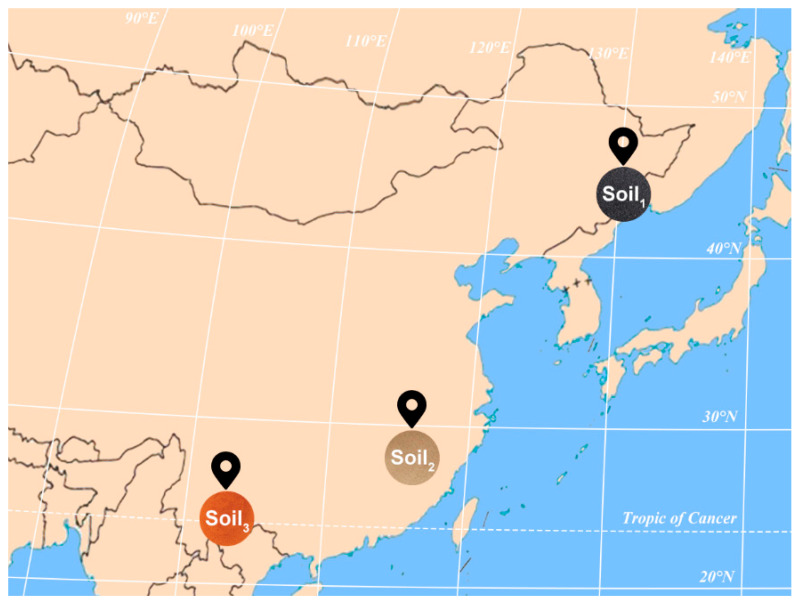
Soil sampling map.

**Figure 2 ijms-23-10404-f002:**
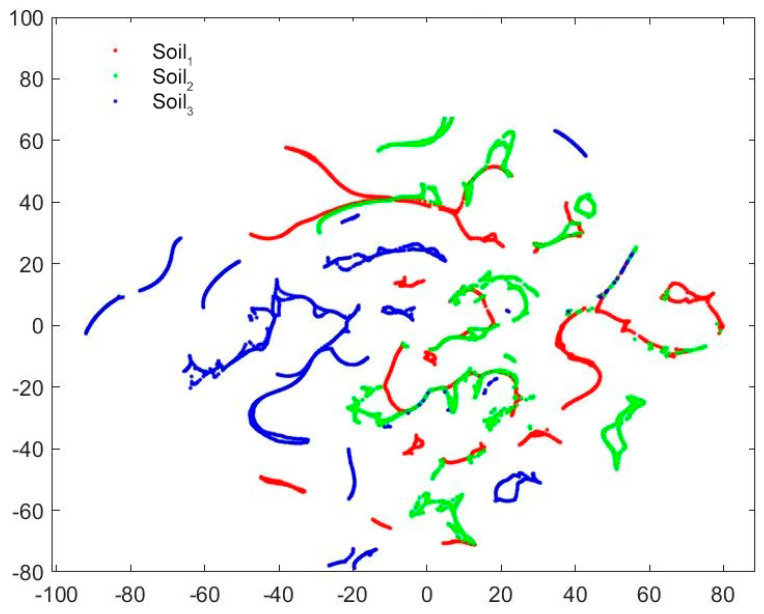
Visualization of three soil samples by t−SNE.

**Figure 3 ijms-23-10404-f003:**
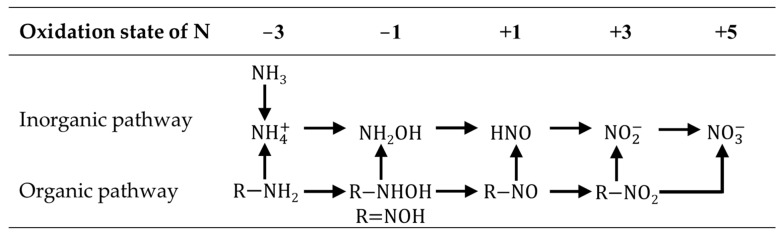
Pathways for nitrification. Adapted with permission from Ref. [29]. Copyright 1977, copyright Plenum Press.

**Figure 4 ijms-23-10404-f004:**
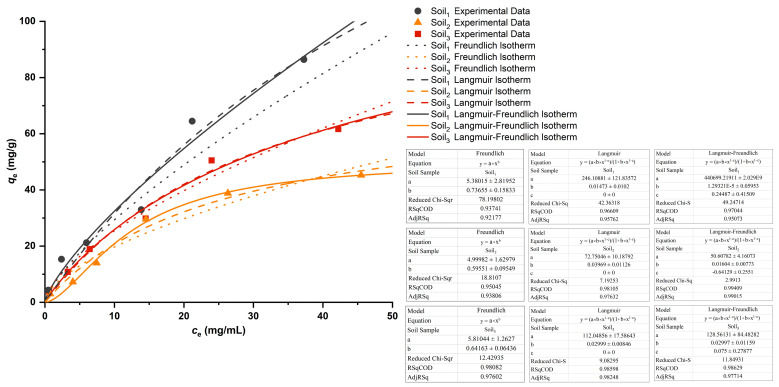
Langmuir and Freundlich adsorption isotherms of the adsorption of ${\mathrm{NH}}_{4}^{+}$ by three soil samples.

**Figure 5 ijms-23-10404-f005:**
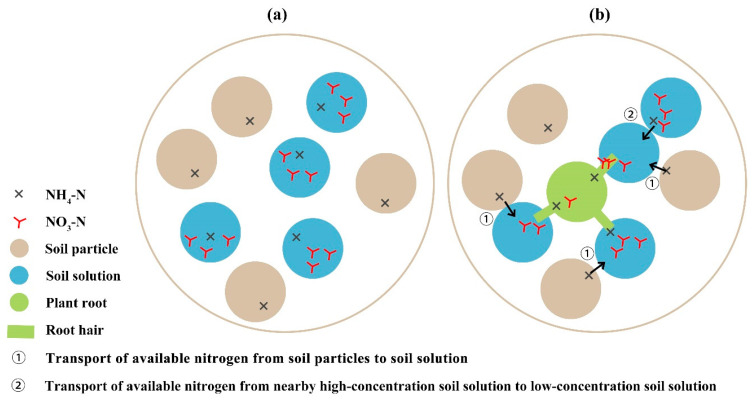
Schematic diagram of transport of soil-available nitrogen to plant roots: (**a**) Initial state of soil-available nitrogen; (**b**) Transport modes of soil-available nitrogen.

**Figure 6 ijms-23-10404-f006:**
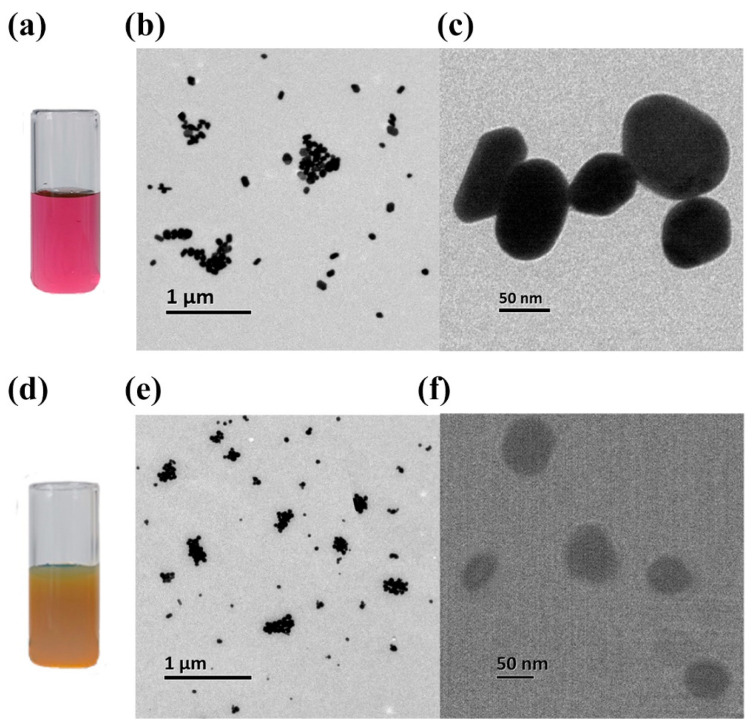
Nanosols and transmission electron microscopy images of their nanoparticles: (**a**) Gold sol in a quartz bottle; (**b**,**c**) transmission electron microscopy images of the gold nanoparticles; (**d**) silver sol in a quartz bottle; (**e**,**f**) transmission electron microscopy images of silver nanoparticles.

**Figure 7 ijms-23-10404-f007:**
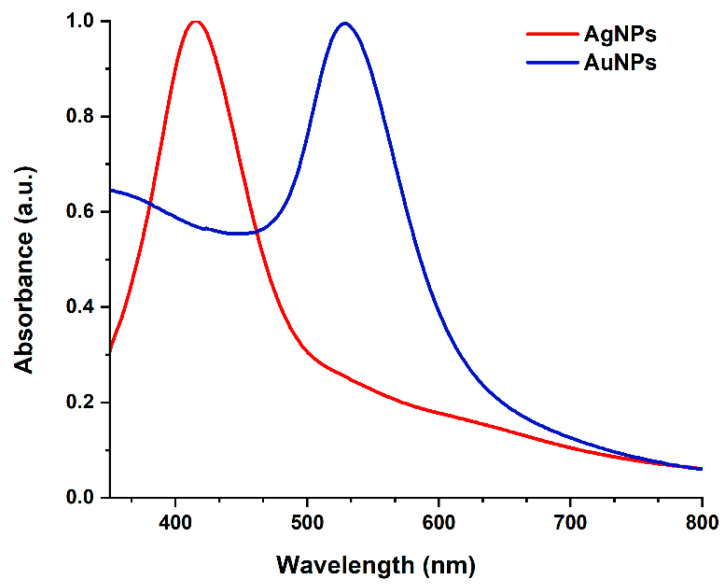
UV-Vis spectroscopy of AgNPs and AuNPs.

**Figure 8 ijms-23-10404-f008:**
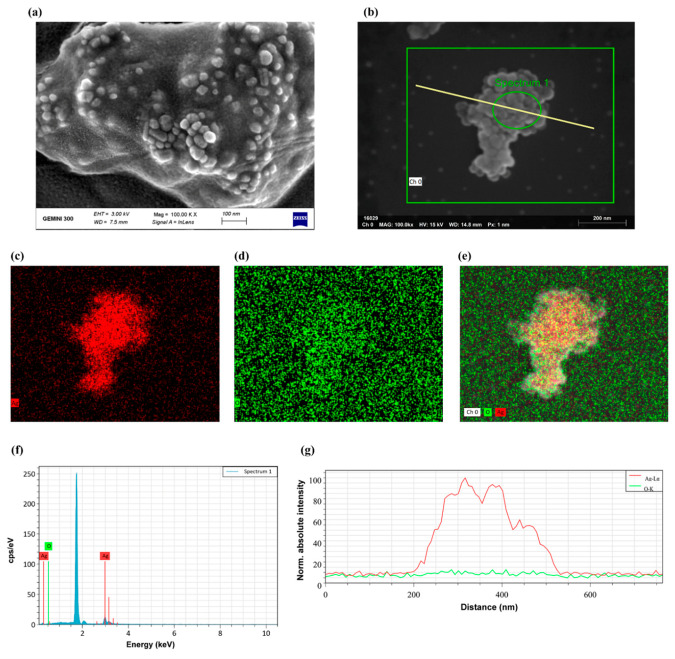
Characterization of silver nanoparticles by scanning electron microscopy: (**a**,**b**) Scanning electron microscopy image of silver nanoparticles; (**c**–**e**) EDS map of sequentially Ag, O and both of silver nanoparticles boxed in (**b**); (**f**) EDS quantification of silver nanoparticles of Spectrum 1 circled in (**b**); (**g**) Normal absolute scattering intensity curves of silver nanoparticles penetrated by the diagonal line in (**b**). All the figures were analyzed by the software (Bruker ESPRIT 2.0) except (**a**).

**Figure 9 ijms-23-10404-f009:**
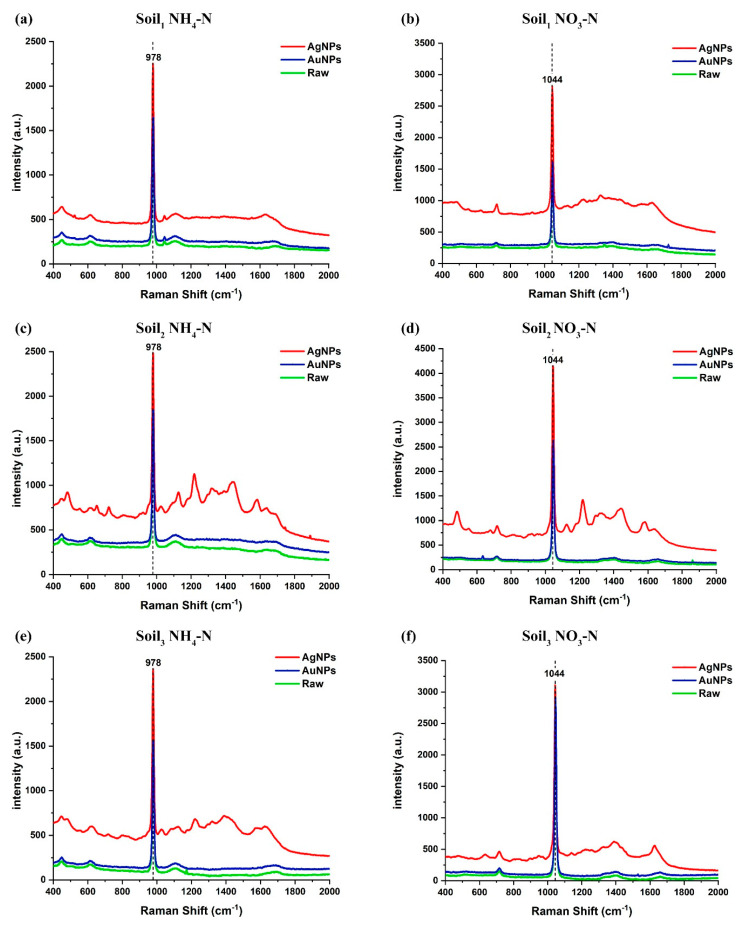
Comparison of Raman spectra of NH_4_-N or NO_3_-N in three soil samples with and without nanosol (with the same nitrogen concentration in the same soil sample): (**a**) Soil_1_ NH_4_-N with nitrogen concentration of 2.377 wt%; (**b**) Soil_1_ NO_3_-N with nitrogen concentration of 1.230 wt%; (**c**) Soil_2_ NH_4_-N with nitrogen concentration of 2.463 wt%; (**d**) Soil_2_ NO_3_-N with nitrogen concentration of 2.157 wt%; (**e**) Soil_3_ NH_4_-N with nitrogen concentration of 2.497 wt%; (**f**) Soil_3_ NO_3_-N with nitrogen concentration of 2.436 wt%. For their legends, Raw denotes the Raman spectra of the raw soil without any addition, AuNPs denotes the Raman spectra measured with the addition of gold sol, and AgNPs denotes the Raman spectra measured with the addition of silver sol.

**Figure 10 ijms-23-10404-f010:**
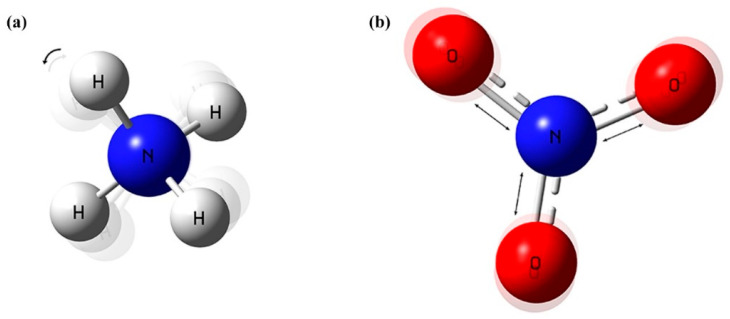
Spatial configuration: (**a**) ${\mathrm{NH}}_{4}^{+}$; (**b**) ${\mathrm{NO}}_{3}^{-}$.

**Figure 11 ijms-23-10404-f011:**
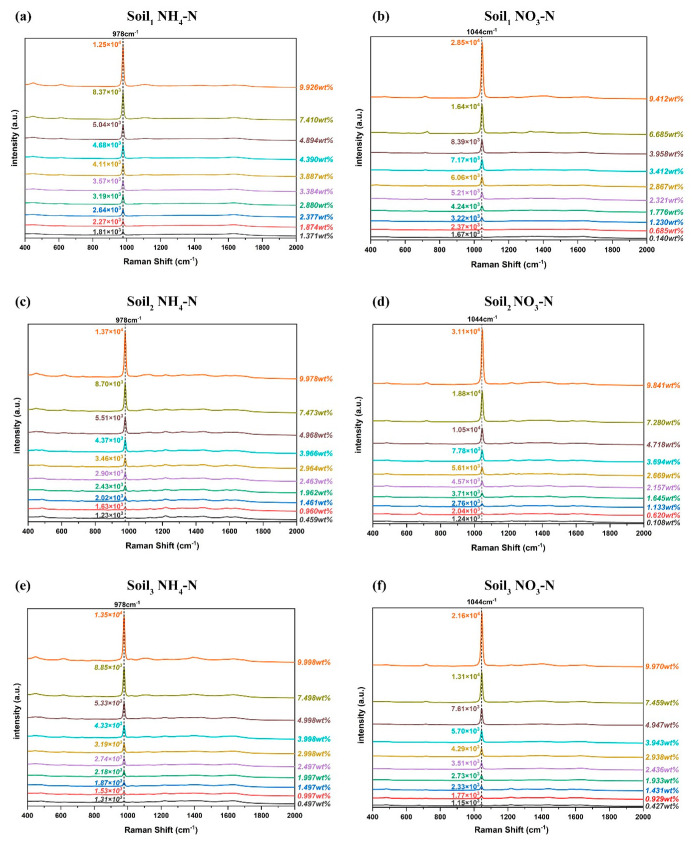
Average Raman spectra of six sample sets: (**a**) Soil_1_ NH_4_-N; (**b**) Soil_1_ NO_3_-N; (**c**) Soil_2_ NH_4_-N; (**d**) Soil_2_ NO_3_-N; (**e**) Soil_3_ NH_4_-N; (**f**) Soil_3_ NO_3_-N.

**Figure 12 ijms-23-10404-f012:**
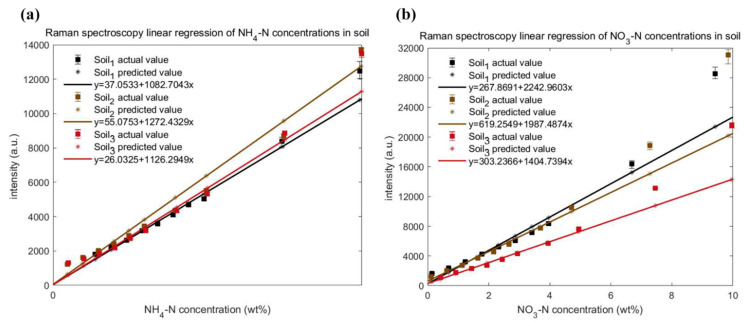
Single variable linear regression visualization for the association between nitrogen concentration and average intensity of Raman peak: (**a**) Each NH_4_-N in three soil samples; (**b**) Each NO_3_-N in three soil samples.

**Figure 13 ijms-23-10404-f013:**
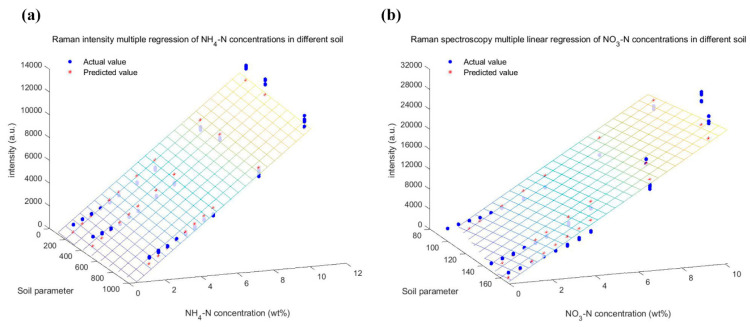
Multiple linear regression visualization for association among soil parameter, nitrogen concentration and intensity of Raman peak: (**a**) All NH_4_-N in three soil samples; (**b**) All NO_3_-N in three soil samples.

**Figure 14 ijms-23-10404-f014:**
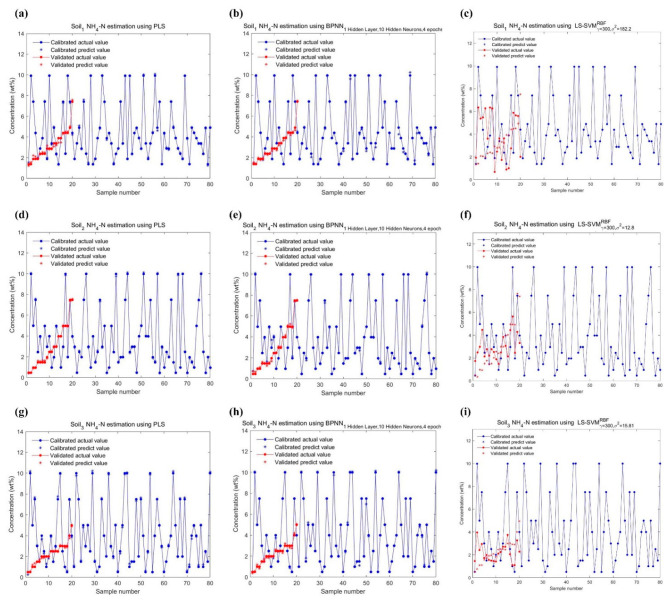
Function estimations of three regression models for NH_4_-N in three soil samples: (**a**) Soil_1_ NH_4_-N estimation using PLS; (**b**) Soil_1_ NH_4_-N estimation using BPNN; (**c**) Soil_1_ NH_4_-N estimation using LSSVM; (**d**) Soil_2_ NH_4_-N estimation using PLS; (**e**) Soil_2_ NH_4_-N estimation using BPNN; (**f**) Soil_2_ NH_4_-N estimation using LSSVM; (**g**) Soil_3_ NH_4_-N estimation using PLS; (**h**) Soil_3_ NH_4_-N estimation using BPNN; (**i**) Soil_3_ NH_4_-N estimation using LSSVM.

**Figure 15 ijms-23-10404-f015:**
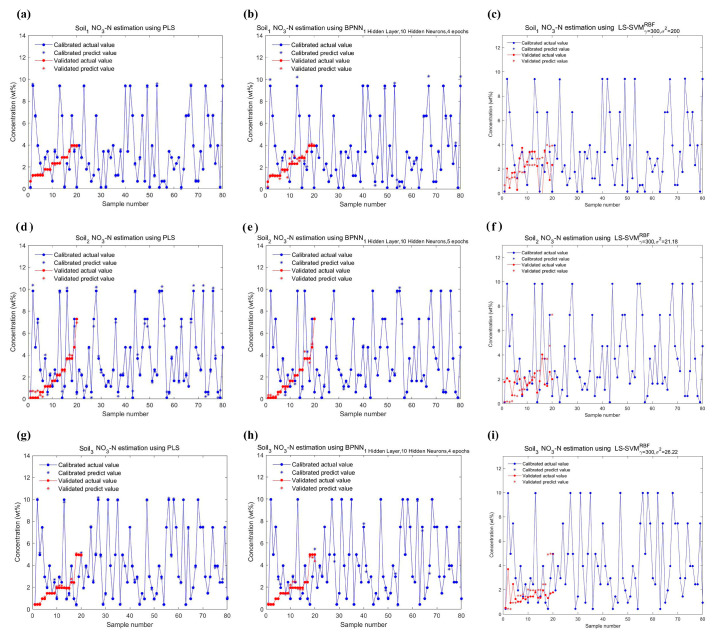
Function estimations of three regression models for NO_3_-N in three soil samples: (**a**) Soil_1_ NO_3_-N estimation using PLS; (**b**) Soil_1_ NO_3_-N estimation using BPNN; (**c**) Soil_1_ NO_3_-N estimation using LSSVM; (**d**) Soil_2_ NO_3_-N estimation using PLS; (**e**) Soil_2_ NO_3_-N estimation using BPNN; (**f**) Soil_2_ NO_3_-N estimation using LSSVM; (**g**) Soil_3_ NO_3_-N estimation using PLS; (**h**) Soil_3_ NO_3_-N estimation using BPNN; (**i**) Soil_3_ NO_3_-N estimation using LSSVM.

**Figure 16 ijms-23-10404-f016:**
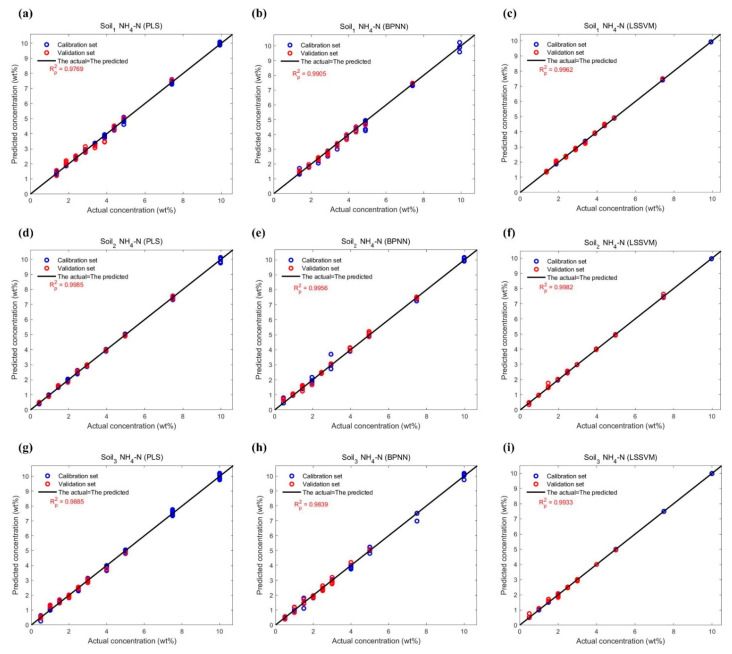
Plots of the actual concentrations of NH_4_-N in three soil samples versus the predicted by three regression models in Figure 14. (**a**) Soil_1_ NH_4_-N estimation using PLS; (**b**) Soil_1_ NH_4_-N estimation using BPNN; (**c**) Soil_1_ NH_4_-N estimation using LSSVM; (**d**) Soil_2_ NH_4_-N estimation using PLS; (**e**) Soil_2_ NH_4_-N estimation using BPNN; (**f**) Soil_2_ NH_4_-N estimation using LSSVM; (**g**) Soil_3_ NH_4_-N estimation using PLS; (**h**) Soil_3_ NH_4_-N estimation using BPNN; (**i**) Soil_3_ NH_4_-N estimation using LSSVM.

**Figure 17 ijms-23-10404-f017:**
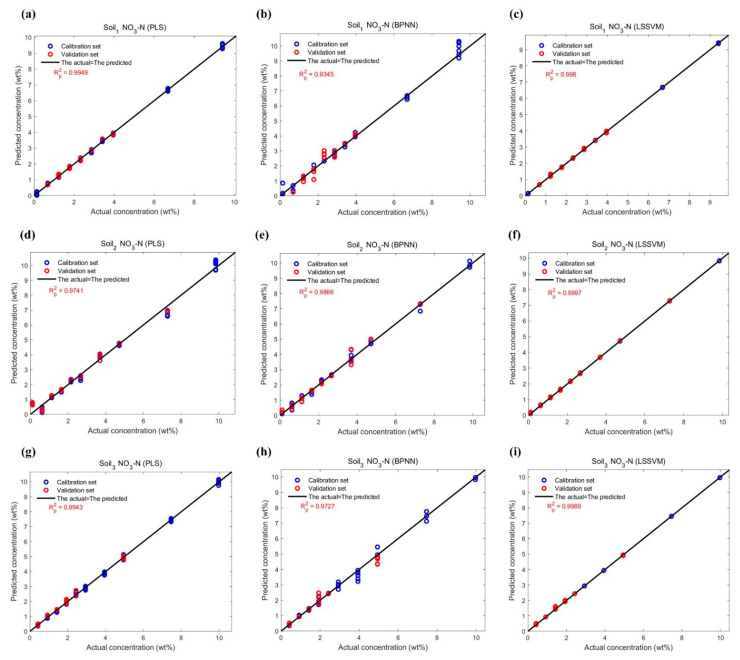
Plots of the actual concentrations of NO_3_-N in three soil samples versus the predicted by three regression models in Figure 15. (**a**) Soil_1_ NO_3_-N estimation using PLS; (**b**) Soil_1_ NO_3_-N estimation using BPNN; (**c**) Soil_1_ NO_3_-N estimation using LSSVM; (**d**) Soil_2_ NO_3_-N estimation using PLS; (**e**) Soil_2_ NO_3_-N estimation using BPNN; (**f**) Soil_2_ NO_3_-N estimation using LSSVM; (**g**) Soil_3_ NO_3_-N estimation using PLS; (**h**) Soil_3_ NO_3_-N estimation using BPNN; (**i**) Soil_3_ NO_3_-N estimation using LSSVM.

**Figure 18 ijms-23-10404-f018:**
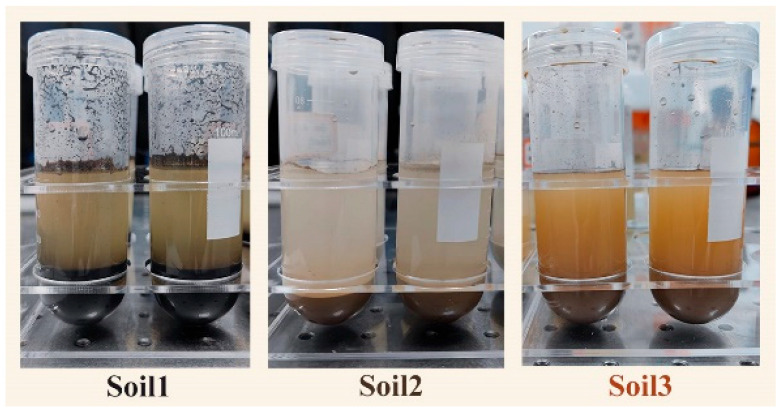
Solutions of three soil samples.

**Figure 19 ijms-23-10404-f019:**
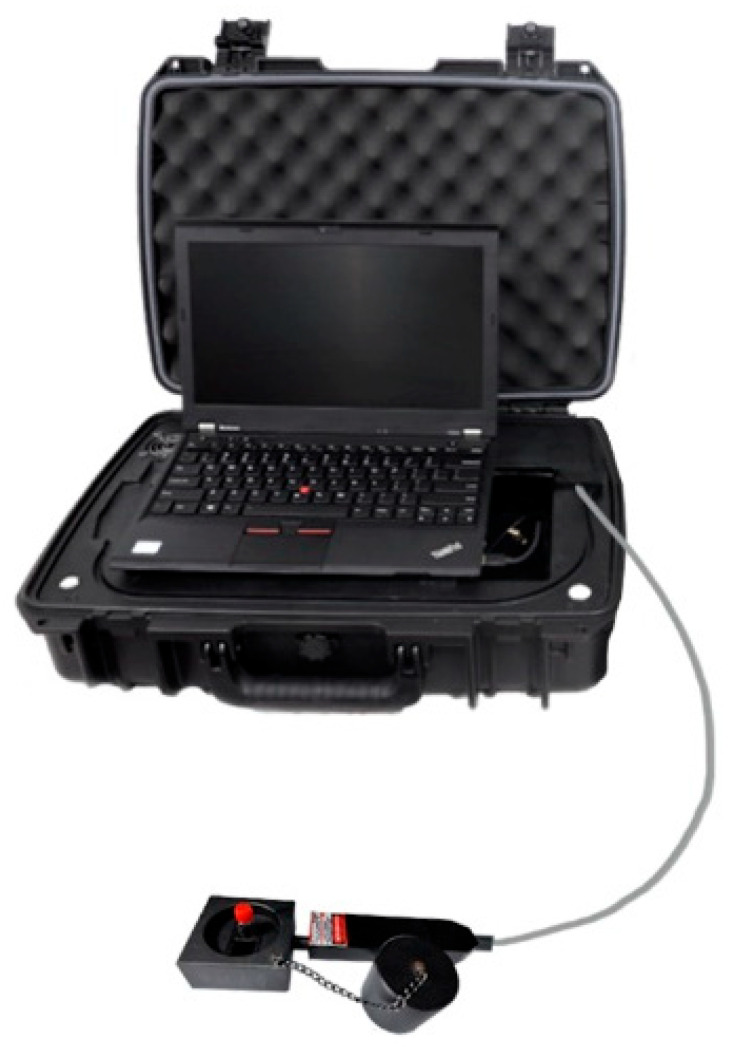
Raman spectrometer with the liquid pool.

**Table 1 ijms-23-10404-t001:** Physical and chemical properties of soil samples.

Soil Sample	pH	Electrical Conductivity (µm/cm)	Organic Matter (%)	Available Nitrogen (mg/kg)	Available Potassium (mg/kg)	Available Phosphorus (mg/kg)
Soil_1_	6.41	25.80	16.63	348.25	5632.87	480.60
Soil_2_	4.70	31.20	2.51	59.06	226.19	212.83
Soil_3_	5.60	15.96	3.50	19.47	80.21	74.34

**Table 2 ijms-23-10404-t002:** Recalculation of pH and electrical conductivity.

	Soil	Soil_1_	Soil_2_	Soil_3_
Index				
pH		3	1	2
Electrical conductivity		2	3	1
pH + electrical conductivity		5	4	3
pH × electrical conductivity		6	3	2

**Table 3 ijms-23-10404-t003:** EDS quantification of elements identified in Figure 8f.

Element	At. No	Net	Atom (%)	Mass (%)	Mass Norm. (%)	Abs. Error (%) (1 Sigma)	Rel. Error (%) (1 Sigma)
O	8	339	34.96	2.28	7.39	0.75	32.82
Ag	47	13,879	65.04	28.64	92.61	0.97	3.38
Total	/	/	100.00	30.92	100.00	/	/

**Table 4 ijms-23-10404-t004:** Performances of six sample sets by single variable linear regression in Figure 12.

Dataset	Sample Set	Single Variable Linear Regression Equation	R_p_^2^	RMSE_r_	RPD_r_
DH_1_	Soil_1_ NH_4_-N	*y =* 37.0533 + 1082.7043*x*	0.984	0.046	5.022
DH_2_	Soil_2_ NH_4_-N	*y =* 55.0753 + 1272.4329*x*	0.974	0.048	6.405
DH_3_	Soil_3_ NH_4_-N	*y =* 26.0325 + 1126.2949*x*	0.975	0.062	4.445
DO_1_	Soil_1_ NO_3_-N	*y =* 267.8691 + 2242.9603*x*	0.959	0.083	2.722
DO_2_	Soil_2_ NO_3_-N	*y =* 619.2549 + 1987.4874*x*	0.963	0.128	1.693
DO_3_	Soil_3_ NO_3_-N	*y =* 303.2366 + 1404.7394*x*	0.963	0.085	1.767

*x*: the concentration of NH_4_-N or NO_3_-N; *y*: the intensity of Raman spectrum.

**Table 5 ijms-23-10404-t005:** Performances of six sample sets by multiple linear regression in Figure 13.

Dataset	Sample Set	Multiple Linear Regression Equation	R_p_^2^	RMSE_r_	RPD_r_
DH	Soil_1_ NH_4_-N	*z* = 124.6083 + 1255.5384*x* − 0.66569*y*	0.9763	0.03903	6.436
	Soil_2_ NH_4_-N				
	Soil_3_ NH_4_-N				
DO	Soil_1_ NO_3_-N	*z* = −6894.454 + 2615.5929*x* + 43.0878*y*	0.9377	0.06098	3.887
	Soil_2_ NO_3_-N				
	Soil_3_ NO_3_-N				

*x*: the concentration of NH_4_-N or NO_3_-N; *y*: soil factor; *z*: the intensity of Raman spectrum.

**Table 6 ijms-23-10404-t006:** Performances of six sample sets by three regression models over the whole wavenumber range.

Dataset	Sample Set	Model	R_c_^2^	RMSE_c_	R_p_^2^	RMSE_p_	RPD
DH_1_	Soil_1_ NH_4_-N	PLS	0.9989	0.0859	0.9769	0.2158	6.6950
		BPNN	0.9973	0.1475	0.9905	0.1429	10.0700
		LSSVM	1.0000	0.0063	0.9962	0.0912	16.0400
DH_2_	Soil_2_ NH_4_-N	PLS	0.9994	0.0712	0.9985	0.0849	24.7400
		BPNN	0.9985	0.1177	0.9956	0.1464	14.4300
		LSSVM	1.0000	0.0040	0.9982	0.0924	22.9500
DH_3_	Soil_3_ NH_4_-N	PLS	0.9980	0.1370	0.9885	0.1356	7.6820
		BPNN	0.9987	0.1125	0.9839	0.1433	7.9710
		LSSVM	1.0000	0.0047	0.9933	0.1023	10.4200
DO_1_	Soil_1_ NO_3_-N	PLS	0.9992	0.0831	0.9949	0.0731	13.6000
		BPNN	0.9951	0.2290	0.9345	0.3055	3.7830
		LSSVM	1.0000	0.0091	0.9980	0.0462	21.6600
DO_2_	Soil_2_ NO_3_-N	PLS	0.9893	0.3197	0.9741	0.3213	5.4110
		BPNN	0.9984	0.1225	0.9866	0.2141	8.7510
		LSSVM	1.0000	0.0045	0.9997	0.0386	47.8000
DO_3_	Soil_3_ NO_3_-N	PLS	0.9988	0.1049	0.9943	0.1147	12.2000
		BPNN	0.9972	0.1625	0.9727	0.2394	5.4510
		LSSVM	1.0000	0.0039	0.9989	0.0560	24.7500

**Table 7 ijms-23-10404-t007:** Sample sets configuration.

Sample Set	Nitrogen Concentration (wt%)	Sample Size
Soil_1_ NH_4_-N	1.371, 1.874, 2.377, 2.880, 3.384, 3.887, 4.390, 4.894, 7.410, 9.926	100
Soil_1_ NO_3_-N	0.140, 0.685, 1.230, 1.776, 2.321, 2.867, 3.412, 3.958, 6.685, 9.412	100
Soil_2_ NH_4_-N	0.459, 0.960, 1.461, 1.962, 2.463, 2.964, 3.966, 4.968, 7.473, 9.978	100
Soil_2_ NO_3_-N	0.108, 0.620, 1.133, 1.645, 2.157, 2.669, 3.694, 4.718, 7.280, 9.841	100
Soil_3_ NH_4_-N	0.497, 0.997, 1.497, 1.997, 2.497, 2.998, 3.998, 4.998, 7.498, 9.998	100
Soil_3_ NO_3_-N	0.427, 0.929, 1.431, 1.933, 2.436, 2.938, 3.943, 4.947, 7.459, 9.970	100

## Data Availability

The data presented in this study are available upon request from the corresponding author.

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
