# Peer review of "Rapid Detection of Available Nitrogen in Soil by Surface-Enhanced Raman Spectroscopy"

_ijms, 2022, doi:10.3390/ijms231810404_

Round 1

Reviewer 1 Report

The article presents the results of rapid detection of Available Nitrogen in Soil by SERS. The article needs major revisions to be published in an impacted journal like IJMS.

1. The article is very interesting, and many text parts are explained straightforwardly. However, the text needs an extensive review of the English syntax. It is sometimes cumbersome to understand or difficult to follow the train of thought.

2. The article is a little messy and confusing. The "materials and methods" section should be put before the "results and discussion" section. The structure of the article should be reviewed. Another example is Figures 5 and 6. In the text, they are described in an inverted way. First, Figure 6 is described, and then Figure 5. Try to avoid this situation when the article is reviewed. 

3. The title. In general, the title is ok. However, we are talking about some experimental results in the laboratory. I think the word "experimental" should be added to the title. 

4. Line 85. I think the authors wanted to see "Raman spectra" instead of "Infrared spectra." 

5. Line 98-102. Is it possible to better explain the methodology by adding the due references? 

6. Line 107. the term "Above" is inadequate in this sentence. 

7. Line 138. The authors mention the presence of "Clay minerals." I agree about this, but how do they detect them? If not detected, is there any reference to the presence of clay minerals in the samples of soil investigated? 

8. Line 142. Clay particles and clay minerals are not direct synonyms. Please, adequate the text according to the real meaning of the words and point 7 of this review report. Please, also add some sentences about the possibility of finding Fe- and Mn- oxy-hydroxides. 

9. Lines 146-160. Please, add the due references. 

10. Figures 2, 12, 13, 14, and 15 are unclear. 

11. Line 178. Add "transport 1 in Figure 3" after symbol 1. 

12. Line 242. Add "respectively" after the LoD indicated. About the values of LoD. The authors report two values that are not directly indicated in the tables and figures. Is it possible to better coordinate all of these data? 

13. Figure 7. What is the concentration of NH4 and NO3 in the reported spectra? It is not clear the spectra of the raw soils. Is it possible to better represent the difference between raw and enhanced spectra? Do the spectra have a baseline correction? Is there any fluorescence in the spectra? In raw spectra, there is a peak between 450 and 500 cm-1. Is it possible the presence of quartz? What are the peaks at ca. 700 and 600 cm-1 shown in many spectra? 

14. Line 268. The authors introduce the blue shift to explain the shift of 2cm-1. This shift is equal to the optical resolution indicated in line 526. It is better to correct the sentence in line 268. 

15. Line 288. The term Soili should become Soili. Please, check and correct all subscripts in the text. 

16. Lines 537-542. Please, add the due references.

17. Line 614. The same as in point 12. 

18. References. Please, I have indicated some parts of the text without references. The chemometrics paragraphs need extensive use of references to strengthen the concepts. 

Reviewer 2 Report

In this study, the authors collected three types of soils (cherno-zem, loess and laterite) and purchased two kinds of nitrogen fertilizers (ammonium sulfate and sodium nitrate) for determination of ammonium nitrogen (NH4-N), nitrate nitrogen (NO3-N) in the soil. The references should be updated to include more recent studies. Their results showed that the portable Raman spectrometer in combination with scatter-enhanced material and machine learning algorithm will be a promising solution for high-efficiency and real-time field detection of soil available nitrogen.

Line 35: To arrange the keywords alphabetically.

Lines 49 – 50: I think that you should add these recent references to support your sentence “Soil N exists in a variety of forms, and soil total nitrogen (TN) encompasses them all, signifying the total storage capacity of soil N.”. I would like to suggest:

Bosso, L., et al.  (2015). Depletion of pentachlorophenol contamination in an agricultural soil treated with Byssochlamys nivea, Scopulariopsis brumptii and urban waste compost: A laboratory microcosm study. Water, Air, & Soil Pollution, 226(6), 1-9.

Keenan, S. W., et al. (2018). Mortality hotspots: nitrogen cycling in forest soils during vertebrate decomposition. Soil Biology and Biochemistry, 121, 165-176.

Lines 82 – 93: Please, describe better your predictions and hypothesis.

Lines 156 – 158: I think that you should add these recent references to support your sentence “As shown in Table1, that is embodied in the differences in pH and conductivity, as well as organic matter and available potassium, etc.”. I would like to suggest:

Bosso, L., et al. (2017). Plant pathogens but not antagonists change in soil fungal communities across a land abandonment gradient in a Mediterranean landscape. Acta Oecologica, 78, 1-6.

Elfanssi, S., et al. (2018). Soil properties and agro-physiological responses of alfalfa (Medicago sativa L.) irrigated by treated domestic wastewater. Agricultural water management, 202, 231-240.

Line 247: Intensity should be written on the y-axis.

Line 278: Intensity should be written on the y-axis.

Line 445: Please, add a map with the countries.

Round 2

Reviewer 1 Report

I sincerely appreciate the improvements that the authors made to the manuscript. The authors have given perfectly satisfactory answers to all my questions. I believe that the manuscript can be accepted in this current form.